# Lithium Biological Action Mechanisms after Ischemic Stroke

**DOI:** 10.3390/life12111680

**Published:** 2022-10-22

**Authors:** Constantin Munteanu, Mariana Rotariu, Marius Turnea, Ligia Gabriela Tătăranu, Gabriela Dogaru, Cristina Popescu, Aura Spînu, Ioana Andone, Elena Valentina Ionescu, Roxana Elena Țucmeanu, Carmen Oprea, Alin Țucmeanu, Carmen Nistor Cseppento, Sînziana Calina Silișteanu, Gelu Onose

**Affiliations:** 1Faculty of Medical Bioengineering, University of Medicine and Pharmacy “Grigore T. Popa” Iași, 700454 Iași, Romania; 2Teaching Emergency Hospital “Bagdasar-Arseni”, 041915 Bucharest, Romania; 3Faculty of Medicine, University of Medicine and Pharmacy “Carol Davila”, 020022 Bucharest, Romania; 4Faculty of Medicine, “Iuliu Hatieganu” University of Medicine and Pharmacy, 400012 Cluj-Napoca, Romania; 5Clinical Rehabilitation Hospital, 400437 Cluj-Napoca, Romania; 6Faculty of Medicine, Ovidius University of Constanta, 900470 Constanta, Romania; 7Balneal and Rehabilitation Sanatorium of Techirghiol, 906100 Techirghiol, Romania; 8Faculty of Medicine, University of Oradea, 410073 Oradea, Romania; 9Faculty of Medicine and Biological Sciences, “Stefan cel Mare” University of Suceava, 720229 Suceava, Romania

**Keywords:** review, lithium, ischemic stroke

## Abstract

Lithium is a source of great scientific interest because although it has such a simple structure, relatively easy-to-analyze chemistry, and well-established physical properties, the plethora of effects on biological systems—which influence numerous cellular and molecular processes through not entirely explained mechanisms of action—generate a mystery that modern science is still trying to decipher. Lithium has multiple effects on neurotransmitter-mediated receptor signaling, ion transport, signaling cascades, hormonal regulation, circadian rhythm, and gene expression. The biochemical mechanisms of lithium action appear to be multifactorial and interrelated with the functioning of several enzymes, hormones, vitamins, and growth and transformation factors. The widespread and chaotic marketing of lithium salts in potions and mineral waters, always at inadequate concentrations for various diseases, has contributed to the general disillusionment with empirical medical hypotheses about the therapeutic role of lithium. Lithium salts were first used therapeutically in 1850 to relieve the symptoms of gout, rheumatism, and kidney stones. In 1949, Cade was credited with discovering the sedative effect of lithium salts in the state of manic agitation, but frequent cases of intoxication accompanied the therapy. In the 1960s, lithium was shown to prevent manic and also depressive recurrences. This prophylactic effect was first demonstrated in an open-label study using the “mirror” method and was later (after 1970) confirmed by several placebo-controlled double-blind studies. Lithium prophylaxis was similarly effective in bipolar and also unipolar patients. In 1967, the therapeutic value of lithemia was determined, included in the range of 0.5–1.5 mEq/L. Recently, new therapeutic perspectives on lithium are connected with improved neurological outcomes after ischemic stroke. The effects of lithium on the development and maintenance of neuroprotection can be divided into two categories: short-term effects and long-term effects. Unfortunately, the existing studies do not fully explain the lithium biological action mechanisms after ischemic stroke.

## 1. Introduction

Stroke is confirmed as the second leading cause of death [1] and the third leading cause of disability [2] worldwide, involving motor, sensory, perceptual, and cognitive dysfunctions [3]. Of the main two types of stroke: hemorrhagic and ischemic, the second is the most frequent type of stroke and accounts for over 80% of all cases [4]. It is characterized by the blockage of blood vessels in the brain. The exact cause of an obstruction of blood vessels in the brain can vary, being attributable to different factors such as plaque buildup in the arteries or embolisms in the heart. Brain injury following obstruction of blood flow can be caused by the impairment of the blood flow in a brain region. Regions with low cerebral blood flow can become irreversibly damaged. Areas within the penumbra with less severe blood flow obstruction are referred to as ischemic penumbra. These regions are considered structurally intact and can function normally. Over time, their characteristics may change and they may develop into an established infarct or a spontaneous recovery [4].

The brain is able to maintain its homeostasis by regulating the transport of water and solute across its various cellular barriers. These include the blood–brain barrier (BBB), the neuroglial cell membranes, and the choroid plexus epithelia. Cerebral edema is a type of cellular pathology that affects the brain’s extracellular and intracellular spaces. The brain contains four different fluid compartments. These include the blood, the cerebrospinal fluid, the interstitial fluid, and the intracellular fluid. In the central nervous system, over 70% of the fluid is composed of the intracellular fluid compartment. The different cellular barriers that surround these four fluid compartments help maintain the proper solute and volume composition of these compartments, which are vital for the functioning of the brain [5]. Ischemic stroke occurs when the blood flow in the brain is reduced due to an obstruction of an artery. This condition usually involves thrombosis or an embolism. The reduction in the flow of blood can lead to a loss of neural function, and this can cause tissue damage [4]. 

The level of potassium and calcium in the fluid is different from that of the interstitial and cerebrospinal fluid. The ion gradients in the brain can also affect the concentration of nutrients in the cells (Figure 1). In order to maintain the concentration of nutrients in the cells, the level of potassium is balanced by the activation of certain energy-dependent ion pumps. These include the Na^+^/K^+^-ATPase and the Ca^2+^-ATPase. The Na^+^-K^+^-ATPase prevents the accumulation of water and solute in the cells, which can lead to cell swelling and the loss of their cytoskeletal integrity. The activity of the Na^+^/K^+^-ATPase can also generate the necessary electrochemical gradients for the transportation of water and ion in the brain [6]. The first fluid compartment that is affected by an ischemic insult is the intracellular fluid. The various energy-dependent processes that regulate the position and volume of the fluid are known to cause cerebral edema. As in other tissues, water is in a thermodynamic equilibrium across the membranes of all brain cells. This means that the osmotic concentration of both extracellular and cytoplasmic fluids is equal [7].

Lithium used at therapeutic doses can exert neuroprotective effects towards the detrimental excitotoxic effects of glutamate [8]. Lithium can stimulate the level of certain cell survival factors, these include the Bcl-2 and the thermal shock protein 70 [9]. It was also shown that lithium can block the activity of the p38 kinase [8], as well as the glutamate-associated JNK activity [10].

By inhibiting glycogen synthase kinase-3β (GSK-3β), lithium influences neurogenesis, exerting multiple actions that are supportive in explaining its therapeutical properties [11]. Upregulation of endothelial Wnt/β-catenin signaling based on GSK-3β inhibition can represent a therapeutic manipulation of BBB integrity as a potential strategy for combating BBB breakdown in the early stages of ischemic stroke [12].

In addition to neuroprotective effects, the inhibition of GSK-3β can also trigger the activation of a long list of transcription factors, such as cyclic AMP response element binding protein (CREB), nuclear factor-κB (NF-κB), activating protein-1 (AP-1), heat-shock factor-1 (HSF-1),β-catenin, T-cell factor (Tcf)/lymphoid enhancer factor (Lef), and p53. CREB, for example, is a transcription factor involved in learning and memory and promotes the expression of Bcl-2 as well as brain-derived neurotrophic factor (BDNF) [13]. By inhibiting the pro-apoptotic effects of so many molecules, lithium was shown to be beneficial. Considering the complexity of the many biochemical processes influenced by lithium, its very narrow therapeutic range and side effects are understandable when not dosed [14].

## 2. Materials and Methods 

This review is based on searching free full-text available papers written in English, found in well-known international databases: Elsevier, PubMed, PMC, ISI—Web of Science, and on Google. In addition, the inclusion criteria were fixed regarding pathology and interventions using specific keywords in the title or abstract: “Lithium” AND “Stroke”. 

## 3. Ischemic Stroke Pathological Context and Lithium Interventions 

In patients with focal ischemic conditions, such as stroke and intracerebral hemorrhage, the presence of thrombin can trigger the formation of gaps in the blood–brain barrier. This condition can also affect the integrity of the vascular endothelium. In addition to this, the presence of vascular angiogenic factor—vascular endothelial growth factor (VEGF)—can also promote the development of vasogenic edema. In addition to these mechanisms, other factors such as the activation of matrix metalloproteinases can also trigger the development of vasogenic edema. When the membrane of the capillary base is damaged, it can lead to the formation of vasogenic edema [15].

Following a stroke, the effects of this condition can lead to the degeneration and death of neurons, loss of motor abilities [16], frequent occurrence of cognitive impairment, depression [1], and changes in the volume of grey matter in the brain—difficult-to-solve issues for treatment and rehabilitation [17]. Excitotoxicity, oxidative stress, and inflammation are the main factors that contribute to this process [18]. The reduction in the volume of grey matter is known to increase the risk of cognitive impairment and dementia [19]. Another common clinical outcome that can result from a stroke is post-stroke depression. There is also a potential link between the volume of grey matter and many other clinical outcomes. Although there is a recovery period window following a stroke, the prevalence of cognitive improvement following a stroke is low [20].

In addition to the pathological changes in the ischemic brain, the presence of additional neuropathological alterations, such as the activation of the hypothalamic–pituitary–adrenal (HPA) axis, can also trigger various physiological changes. The presence of these alterations can contribute to the development of brain injury by affecting the various aspects of stroke progression. These abnormalities are often reflected in the multi-system complications that can affect the clinical course of a stroke [21]. 

Currently, the only available and approved treatments for post-stroke sequelae are acute thrombolysis [22] and mechanical thrombectomy [23]. Lithium may reverse the grey matter atrophy by stimulating neurogenesis [24]. Several neuroprotective agents, including lithium, have been shown to stabilize the blood–brain barrier (BBB) as part of their neuroprotective mechanisms [25]. The endothelium is a critical component of the BBB function, and it plays a role in regulating its properties. The Wnt/catenin signaling is responsible for the regulation of the barrier’s properties. Lithium is known to upregulate the activity of the β-catenin gene by inhibiting the GSK-3β enzyme [26]. 

Lithium promotes acute post-stroke neuronal survival [27]. Lithium has been shown to promote neurogenesis [28], protect against cell apoptosis through autophagy induction [29], and increase the global volume of grey matter [30]. These features make it an ideal candidate for further studies on the effects of lithium on cognitive recovery following a stroke and other neurological disorders. A meta-analysis of the data collected from bipolar populations revealed that there is a positive correlation between lithium use and the volume of grey matter. The effectiveness of lithium in stroke also involves lithium–endothelium interactions [31] by improving vascular or cerebrovascular autoregulation of blood flow, dynamic endothelial barrier permeability, and vasorelaxation capacity [32].

Hypothermia can provide various benefits in rehabilitation and neurocritical care [4]. Hypothermia treatment can affect the regulation of the expression of the insulin-like growth factor 1 gene [33]. It has been known that lithium chloride can prevent hypothermia by inhibiting the Tau hyperphosphorylation and enhancing the expression of catenin protein. The body temperature of patients following an ischemic brain stroke can increase significantly in the following 6 to 72 h [34]. This phenomenon is known to be an indicator of the outcome of the disease. In vitro studies suggest that hypothermia can be used to control the expression of the insulin-like growth factor 1 gene [35].

In cultures, the presence of lithium significantly increased the number of stem cells as well as their proliferation and differentiation. It also reduced the number of macrophages and microglia. Lithium has dual effects on the differentiation and proliferation of stem cells in the central nervous system. It is believed that lithium activates the receptors of the BDNF [36]. This suggests that the activation of this receptor could be a signal that promotes the growth of the stem cells. The effects of lithium on the generation of neurons by the hippocampal neural stem cells have been studied. The presence of BDNF significantly increases the number of neurons that are derived from this stem cell. Several studies also suggest that the presence of this protein in the precursor cells can hamper the survival and proliferation of these cells. Other molecular pathways that could play a role in the development of neurons include the activation of the ERK/Bresor and GSK-3 inhibition. It was also confirmed that the use of lithium does not alter the level of glial differentiation [24]. 

## 4. Data on Lithium Biological Action Mechanisms after Ischemic Stroke

Lithium, discovered in 1817 by Arfvedson, is found in tiny amounts in all soil types. It is especially prominent in the luteic (clay) fraction and less so in the organic soil fraction, and appears in amounts between 7 and 200 μg/g. Lithium concentrations in groundwater reach up to 500 μg/L. In some natural mineral waters, a lithium concentration of up to 8 mg/L is reached [11]. In some regions, drinking water also provides significant amounts of the element [37]. Lithium is also found in varying amounts in food: the main food sources are grains and vegetables. The intake of lithium in the human diet depends on the location and type of food consumed and varies in quantity between 650 and 3100 μg [11]. The basal serum level of lithium in adults is in the range of 7 to 28 μg/L, corresponding to daily lithium intakes of 385–1540 μg. The concentration of lithium in the hair reflects the average intake of available lithium over time and is a non-invasive way to determine the required amount of lithium in the diet [11].

Lithium was initially used to treat depression during the 19th century. In 1949, the antimanic effects of lithium were confirmed through a double-blind study. In the 1960s, it was discovered that lithium could prevent the recurrence of depression and mania [1]. During the 1970s, significant controversy emerged regarding the safety of lithium. It was reported that long-term use of lithium could cause kidney damage. However, a multicenter study by international researchers revealed that long-term lithium treatment could lower suicide and mortality rates, but also can reduce the risk of stroke [38]. 

In the form of soluble salts, lithium is absorbed by the small intestine through the Na^+^ channels. It is then transported to the kidneys and is completely degradable. The level of lithium in body water is similar to that of the extracellular and internal levels. The levels of lithium in the serum are influenced by the amount of lithium that an individual takes [39]. Adults taking 0.25 mM (1.74 mg) of lithium as chloride per day for several weeks express a mean serum lithium concentrations increased from a baseline concentration of lithium [40].

Lithium can be transported through membranes in five different ways. Of these, passive flow is the most important route of entry for lithium into cells, and co-transport of sodium-lithium for the expulsion of lithium from cells. The five components of Li^+^ transport are: 1. The Na^+^-dependent antiport transport system transports Li^+^ in both directions through the plasma membrane; 2. The Na^+^/K^+^ pump mediates the uptake of Li^+^ but not its release from cells with physiological Na^+^ and K^+^ content. Both Na^+^ and external K^+^ inhibit the uptake of Li^+^ via the pump into choline environments; 3. Li^+^ can enter the cell through the voltage-dependent Na^+^ channel. The entry of lithium in this way is stimulated with veratridine and scorpion toxin, the stimulation being blocked by tetrodotoxin; 4. The residual pathways comprise a saturable component, which is comparable to the basal Na^+^ uptake; and 5. An ouabain-resistant component that causes an expulsion of Li^+^ against the electrochemical gradient in choline environments [11].

Lithium has a simple structure, and its bioactive properties have been known for over a hundred years. It has been used to treat bipolar disorder, also known as manic-depressive illness. It has also been used to treat various mental disorders like depression and anxiety. Studies have shown that lithium can protect nerve cells from multiple attacks, such as toxins, stress, ischemia, and injury [41]. In addition to its mood-stabilizing effects, lithium has other physiological properties that can affect the development of various tissues. Lithium has been shown to exert a pleiotropic neuroprotective effect on different neurological conditions by triggering various pro-survival mechanisms. These mechanisms mainly affect the regulation of inflammation, neurogenesis, and apoptosis [42].

Both in vitro and in vivo data revealed that lithium exerts various effects on neurotransmitters [43] and receptor-mediated signaling regulation. In addition, data showed that lithium could influence the development of gene expression and hormonal regulation. Although the molecular mechanisms by which lithium exerts all its effects are still unknown, its various effects are believed to be linked to the multiple interactions with molecular signaling components. Lithium’s acute effects are mainly mediated by inhibiting certain enzymes involved in the regulation of two signaling pathways. These include the protein kinase C and the glycogen synthase kinase 3β [13]. During chronic lithium administration, the expression of different genes is severely affected. Chronic lithium administration significantly increases the expression of a neuroprotective protein known as Bcl2, with some of the lithium effects being mediated by its neurotrophic properties [44].

In 1971, it was discovered that the levels of inositol in rats were reduced by lithium. Due to its neurotropic properties, lithium is now regarded as a promising drug for treating various diseases [45]. It is also known that it can block the activity of certain enzymes that are involved in the regulation of the myo-inositol monophosphate cycle and glycogen synthase kinase 3β. The same activity is related to lithium’s influences on diabetes and cancer [11]. 

In addition to being useful in treating bipolar disorder, lithium can also be used to prevent the development of other processes, such as hematopoiesis and the synthesis of glucose. Several enzymes are known to be potential targets of lithium’s action. Various studies suggest that the effects of lithium on the signaling pathways that are involved in the development of complex physiological and behavioral traits can be inhibited by lithium. These include the cyclic AMP formation, the G protein, and the inositol phosphate metabolism [11].

Lithium is known to be an effective inhibitor of the multiple structurally related magnesium-dependent phosphomonoesterases. It can be found in concentrations that are therapeutically relevant for these enzymes. The G protein family is a ubiquitous group of proteins that play a critical role in the regulation of various physiological and behavioral traits. These include the development of mood and appetite. The first direct evidence that lithium can be used as a target of its actions was provided by a study conducted by Avissar and colleagues, which revealed that the presence of lithium significantly decreased the binding of isoproterenol and carbochol to various G-proteins in the cerebral cortex of rats [11].

Lithium is known to stimulate neurogenesis and exert neuroprotective effects by inhibiting glycogen synthase kinase-3 (GSK-3β) [46]. It also modulates the activity of various neuroprotective factors, such as B-cell lymphoma-2 and heat shock protein 70. It can also downregulate pro-apoptotic factors. Lithium has been shown to reduce the death of neurons, activate the cyclooxygenase-2 pathway, and maintain BBB integrity [9]. In addition, the presence of GSK-3β can also cause the activation of certain nuclear factors. For instance, it can phosphorylate the Wnt/β-catenin and the nuclear factor of activated T cells. Inhibiting the activity of GSK-3β can also influence various other transcription factors, such as the AMP-activated protein and the mitogen-activated protein [9]. 

Studies have shown that chronic lithium treatment at therapeutic doses provided complete protection against the effects of glutamate-induced excitotoxicity in various types of neurons [47,48]. These neuroprotective effects were independent of the activity of inositol monophosphatase. However, they were associated with the downregulation of the calcium influx through the NMDA receptor. It is not yet clear if this aspect of lithium’s action requires GSK-3β inhibition [9].

Lithium is also able to block the pro-apoptotic effects of certain molecules, such as p53 and Bax [49]. It inhibits the activation of caspase-3 [50] and the cleavage of laminin B1 [49]. In addition, by increasing the activity of the serine-threonine Akt-1 kinase, lithium can also prevent the development of neurons that are prone to apoptosis. For instance, by increasing the levels of phosphorylated and active Akt-1 in the cerebellar neurons, they were able to respond to higher levels of toxic chemicals [51].

The presence of lithium significantly increased the activity of the PI3K and the kinase activity, which reversed the loss of the latter’s activity and cell viability. In addition, it was shown that the treatment of patients with acute lithium exposure protected cortical neurons from the effects of glutamate [9].

It has been known that lithium can increase the levels of nerve growth factor (NGF) in the hippocampus, amygdala, and the frontal cortex. However, no effects were detected on the levels of NGF in the hypothalamus, striatum, or midbrain [9].

In an analysis of the effects of lithium on the expression of VEGF, it was revealed that the drug promoted the development of vascular cells through a pathway that is independent of the PI3K/GSK-3 receptor. These findings support the idea that the neuroprotective effects of lithium are partially mediated by the activation of the VEGF [9].

A new member of the FGF superfamily, fibroblast growth factor-21 is known to regulate the development of fat and glucose metabolism. It is also known to target the liver and pancreatic islets. It is a potential therapeutic target for diseases such as diabetes and obesity [9].

Numerous studies have shown that lithium can induce the development of neurogenesis. In a rodent model of striatal injury, lithium exposure induced neurogenesis by increasing the number of neurons and their phenotypes [9]. However, in older rats, the effects of lithium on the development of neurogenesis were not as significant [9]. Chronic lithium treatment significantly increased the differentiation of neural stem cells in the hippocampus. In studies on human brain cells, lithium was shown to stimulate the development of nestin-positive stem cells [9]. The effects of lithium on the development of neurogenesis were shown to be significant when it came to increasing the number of neurons that were derived from transplanted cells. It was also shown that the drug inhibited the activation of the macrophages and the PI3K/Akt pathway. In addition, it was revealed that lithium prevented the development of the Notch gene in Chinese hamster oocytes [9].

Due to the properties of lithium and its multiple mechanisms of action, it has been used to investigate the possible neuroprotective effects of this drug on various conditions. Some of these include Parkinson’s disease [52], Alzheimer’s disease, and Down syndrome. In addition, it has been shown that lithium can improve the cognitive performance of patients following a stroke [9].

In studies on mice, lithium was shown to reduce the volume of lesions following injury induction [53]. It was also able to prevent the development of neuroinflammation caused by the activation of the cyclooxygenase-2 and the activation of the microglial. In addition, it was able to maintain the integrity of the blood-brain barrier by inhibiting the expression of the matrix metallopeptidase-9 [9].

In studies on mice, lithium was shown to reduce the levels of amyloid precursor protein (APP) in the brains following injury [54]. It was also able to prevent the development of neuroinflammation caused by the activation of the matrix metalloproteinase [55]. In studies on the effects of lithium on spatial learning and memory, it was discovered that it can reduce the loss of hippocampal volume caused by brain injury [56]. This evidence supports the idea that lithium can prevent the development of negative downstream effects, such as memory deficits. It also shows that the drug can reduce the levels of tau phosphorylation and APP accumulation [9].

In animal models of stroke, the upregulation of GSK-3β was revealed to be a neuroprotective mechanism. This mechanism was also linked to the induction of BDNF, which plays a role in the development and maintenance of neurons. In addition, the increased expression of VEGF in the vascular endothelium was found to induce neurovascular remodeling. The inhibition of the N-methyl-D-aspartate receptor, which is a critical factor in the development and maintenance of neurons, was shown to decrease glutamate excitotoxicity. It also inhibited the regulation of autophagy, which is a process that contributes to the survival and functioning of neurons [57].

Metalloproteinases play a role in the development and maintenance of atherosclerosis by participating in the inflammatory process and the degradation of the extracellular matrix. When the production of this metalloproteinase by macrophages is restricted, it can lead to the formation of a thrombus. By suppressing the production of this metalloproteinase, lithium treatment was found to reduce the level of this enzyme. The development and maintenance of atherosclerosis can also be influenced by the presence of vascular smooth muscle cells (VSMCs), known to play a role in the proliferation and migration of the arterial wall. They are also involved in the abnormal neovascularization of the plaque. In a study, lithium inhibited the migration and proliferation of VSMCs in an animal model. It also reduced the cholesterol and blood glucose levels in the animals. These findings suggest that lithium can help decrease the detrimental effects of hyperlipidemia and diabetes on stroke events [58]. The above mechanisms may contribute to the beneficial effects of lithium on cerebral ischemia and atherosclerosis. In a study, lithium was shown to reduce the accumulation of atherosclerotic plaque in mice [59]. Another study revealed that the combination of lithium and captopril reduced the risk of stroke and improved the survival of rats that were prone to experiencing severe stroke [59].

Doeppner et al., examined lithium in a rat MCA occlusion model. They noted that when lithium was administered within 6 h of onset, reduced infarct volumes, edema, leukocyte infiltration, and microglia activation were present [27]. They reported that lithium increased levels of miR-124, resulting in the degradation of the RE1-silencing transcription factor and thus leading to postischemic neuroplasticity. This effect was independent of glycogen synthase kinase 3β (GSK3β) [60]. 

Lithium is a non-selective inhibitor of the glycogen synthase kinase-3β. It can also stimulate neurogenesis, promote the production of neurotrophins, and prevent post-injury inflammation. The imbalance between the production of reactive oxygen and the effectiveness of the antioxidant system can lead to detrimental effects on cells. The downstream target of Nrf-2 and other antioxidative enzymes, such as glutathione and superoxide dismutase, play a significant role in protecting cells from the effects of oxidative stress. They also regulate the levels of antioxidants and cytoprotective genes. Nrf-2 is a component of the cell’s antioxidant system that is associated with Kelch-like ECH protein 1. Upon stimulation, it is transferred to the nucleus, where it plays a role in the transcription of various antioxidants and phase II genes [61]. 

The effects of lithium on the activity of phospholipase A2 in the brain were also studied. They revealed that chronic lithium administration significantly decreased the turnover of arachidonic acid in various brain phospholipids. In addition, this effect can also block the activity of the brain PLA2 in signal transduction. The study also revealed that chronic treatment with lithium significantly increased the levels of protein and mRNA for calmodulin, which are both sensitive and insensitive to the drug. It was also observed that the presence of lithium in rats also decreased the levels of Gi2 and Gi1. Although the effects of lithium on the two G-proteins were not seen in response to short-term treatment, they were observed in response to chronic treatment. The presence of chronic lithium has been known to affect the activity of the protein kinase C and the glycogen synthase-3β in signal transduction pathways. These two are known to be involved in the regulation of synaptic function. By linking the expression of these two genes to the actions of lithium, a strategy can be developed to identify the signature genes that are involved in lithium treatment [62].

The complex effects of lithium on the development and maintenance of neuroprotection are known to be important for its therapeutic effect. These include its ability to stabilize the activities of the neurons and support neural plasticity. It is also believed that the effects of lithium on the modulation of neurotransmitters can contribute to neuroprotection [63]. Lithium also affects the signals that are delivered to the cytoskeleton, which are known to contribute to neural plasticity. These include the AMP-dependent kinase, the protein kinase C, and the glycogen synthase-3β. These three kinases are known to be involved in the regulation of mood recovery and stabilization. Lithium achieves its antiapoptotic function through the protein-mediated signaling pathway Kinase B (Akt). Akt is a serine-threonine kinase and a proto-oncogene that has a phospholipid-binding domain used to anchor them to the plasma membrane [11].

Although it is believed that lithium can replace sodium in the sodium-transport system, the biological significance of this process is still unclear. It is currently not possible to determine the concentration of lithium in excitable cells. Due to the presence of certain ion channels that can easily accept lithium, it has been theorized that there should be a tenfold increase in the concentration of this chemical in excitable cells. In order to maintain its normal potential, this concept should be examined by taking into account the Nernst equation. Lithium concentration in cells does not correspond to the Nernst equation’s prediction, instead, it is lower than the concentration in extracellular and blood fluids [39].

Excitotoxicity is a process that can contribute to the development of atrophy and damage to neurons following an ischemic event [37]. Neurons require oxygen to produce adenosine triphosphate, which is a vital component of the brain’s energy supply. During an ischemic event, the lack of oxygen can prevent the production of new ATP. The Na^+^/K^+^-ATPase is a vital component of the brain’s energy supply and is involved in the ionic gradient across the cell membranes. During an ischemic event, the lack of oxygen can prevent the production of new ATP. The proper function of this enzyme is required to prevent the unwanted depolarization of the cell. Due to the limited number of ATP stores in the brain, the transport system is unable to maintain cell membranes’ ionic gradient. The reduction of the number of neurons’ excitatory neurotransmitters can lead to the release of these chemicals. In addition to this, the lack of reuptake can also lead to the over-expression of certain chemicals in the neurotransmitters. These include glutamate, which can cause the activation of certain post-synaptic receptors [63]. Over-expression of these receptors can lead to the activation of a certain type of cell-damaging protein known as the Ca^2+^ influx. This process can cause cell structure to be destroyed through the degradation of the cell’s calcium-dependent protein kinases. Although this process is considered to be the main cause of infarcts during the acute phase, its role in the development of other conditions is not clear [63].

Two types of non-oxidative substances (NOSs) known as calcium-sensitive forms are commonly activated in neurons when exposed to high levels of Ca^2+^. Following an ischemic insult, the activity of these two types of compounds is increased, which can lead to the development of more toxic chemicals such as superoxide and nitric oxide. In addition to this, a variety of other processes can also affect the development of superoxide. One of these is the activation of the xanthine dehydrogenation enzyme, which is carried out through the mitochondria’s electron transport chain activity [57].

When superoxide is combined with nitric oxide, it forms peroxynitrite, which can cause DNA damage. This process can trigger DNA repair mechanisms, which are known to be involved in the development of post-ischemia apoptosis in models. In addition to this, the presence of reactive oxygen species can also affect the proton gradient across the cell membranes. This can cause the release of signals that are related to cell apoptosis. There is also evidence linking lithium treatment to the pyroptosis mechanisms and oxidative stress, through the Nrf2-HO-1 pathway [64].

The inflammatory response to an acute cerebral infarction is a dual mechanism, detrimental, but also a protective factor that can determine the outcome of the stroke. Activation of various immune cells such as B cells, T cells, and macrophages can help in reducing the damage caused by an ischemic stroke [65]. In addition, cerebral ischemia can trigger the production of cytokines and neutrophil infiltration. The c-Jun N-terminal protein kinases (JNKs) are involved in the regulation of various processes, such as the formation of non-transcriptional and inflammatory responses, as well as the activation of the apoptosis and necrosis signaling pathways. Li^+^ has neuroprotective properties for these threats [66].

The effects of Li on inflammation have been investigated. Fewer cytokine-secreting cells (interleukin [IL]-6, IL-10, IL-2, and interferon [IFN]-γ) were found in bipolar patients after chronic Li treatment, suggesting that Li may normalize immune activation [67]. When activated, the presence of oxidizing stress can stimulate the development of an inflammatory cascade in the parenchyma. This process can trigger the release of various types of inflammatory mediators, such as chemokines and cytokines. Dedicated studies on the development of these conditions are focused on the treatment of tumor necrosis factor α and IL-1 [68]. IL-1 is known to be a proinflammatory cytokine that can be involved in the development of post-ischemia damage in the brain. In studies, it has been shown that inhibiting or knocking out the activity of this cytokine can reduce the number of neurons injured in the reperfusion phase [69]. In addition to this, in vitro studies suggest that it can also induce cell apoptosis through the activation of nitric oxide-induced processes [70]. It has been theorized that tumor necrosis factor can regulate the activity of apoptosis following an ischemic insult. In studies, it has been shown that knockout animals that were treated with tumor necrosis factor showed better survival compared to those that were not treated [71]. Other studies suggest that this factor can also be involved in the restructuring and neuroprotective processes following stroke [72]. 

The inflammasome, which is a component of the central nervous system’s response known to intervene in the development and progression of ischemic stroke. The function of the inflammasome is to recognize various danger signals that are associated with the development of lesions in the nervous system. Existing theories suggest that the interaction between the inflammasome and various other molecules, such as lysosomal rupture and potassium efflux, can trigger the accumulation of reactive oxygen species (ROS) [1]. 

Autophagy is a regulated process that involves the recycling and degradation of damaged macromolecules to maintain cellular homeostasis [73]. In neurons, the regulation of the synthesis and degradation of protein is very important for the development of synaptic plasticity and cell growth [74]. Various studies have shown that lithium can promote autophagy activation in different central nervous system cells [75]. These include neurons, astrocytes, macrophages, and capillary endothelial cells [31] upon an ischemic event. Autophagy can play a dual role related to the removal of damaged or senescent cells and in the protection of the brain following an injury [76] by preventing the development of downstream apoptosis. This is because the degradation of damaged mitochondria through mitophagy can help prevent downstream apoptosis. Unfortunately, excessive autophagy can lead to the destruction of various cellular functions. It can also be detrimental to neurons, which eventually leads to cell death. In addition, evidence has shown that the activation of cerebral IR could cause this process to negatively affect the neurons. It is known that prolonged cell stress can trigger excessive activation of the autophagy process. This could explain why the reduction of this process in the presence of chronic IR-induced hippocampal neuronal damage can help alleviate cognitive impairment [74]. One way to prevent the development of downstream apoptosis following an injury is by using lithium chloride [2]. 

Apoptosis can be mediated through the activation of various pathways, such as those that are dependent on the presence of a certain type of protein known as caspase-3. In neurons, the activation of this protein can be triggered by the release of a certain type of chemical by the mitochondria. This process then leads to the breakdown of the structural proteins. It is not clear how the activation of the JNK and AIF can lead to cell death. However, it is believed that the activation of these two factors can cause structural and DNA damage that ultimately leads to cell death. In order to prevent the proliferative process from taking place, certain B cells can be kept in check. These include B-cell lymphoma 2 and nuclear factor NF-kB [11].

Circadian (daily) rhythms exist in all eukaryotes and are conducted by endogenous, self-sustaining biological oscillators; in the absence of external time interventions, the rhythm will last for almost 24 h. The effects of lithium under synchronous conditions are specifically variable, with significant delays or phase advances. The effects also depend on the time of its application, so the administration of lithium hydroxybutyrate in the evening for 10 days stabilizes circadian rhythms in rats, in contrast to an administration during the morning. Inositol phospho-metabolism has been proposed as a potential component of both the circadian oscillator and blue light photo-transduction [11]. The therapeutic action of lithium salts appears to be the result of a combination of events that alter neuronal activity at multiple levels (Figure 2). Three interacting mechanisms appear to be more affected [11]:

(1) The modulation of neurotransmitters by lithium ions seems to change the ratio between their excitatory and inhibitory activities and the decrease of glutamatergic activity may contribute to the effect of neuroprotective lithium [77];

(2) Lithium ions modulate signals with an impact on the cytoskeleton, a dynamic system that contributes to multilevel neuronal plasticity, including glycogen synthetase kinase-3β, cAMP-dependent protein kinase, and protein kinase C;

(3) Lithium ions alter signaling activities involving secondary messengers, transcription factors, and therefore gene expression.

**Figure 2 life-12-01680-f002:**
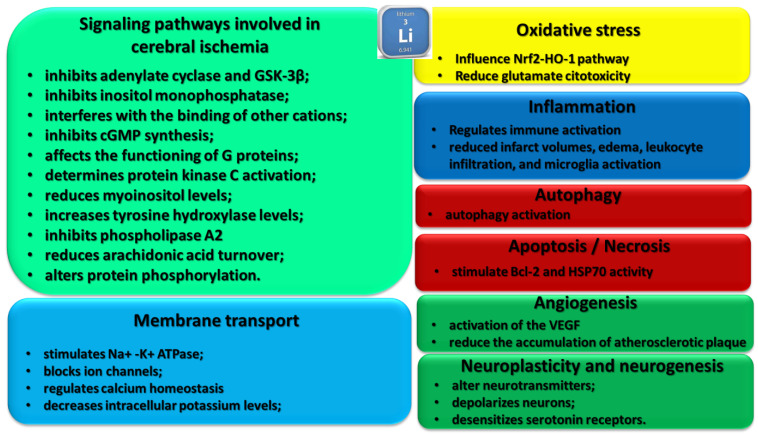
Synthetic overview of the lithium effects in various cellular functions correlated with possible therapeutic interventions in ischemic stroke, such as membrane transport, signaling pathways, neuroplasticity, neurogenesis, angigenesis, apoptosis, necrosis, autophagy, inflammation, and oxidative stress.

## 5. Discussion

The properties of lithium are well-characterized. It is an 80–100% bioavailable compound, and it can be administered orally. It can reach its peak serum concentration in around two to four hours. It is also water-soluble, which makes it ideal for dissolving in water [11]. Although the effects of lithium on mood are known to be beneficial, it is also known to have side effects such as polyuria and polydipsia [78]. These side effects can be caused by the lack of thyroid function. It is also believed that the side effects of lithium are caused by the actions of its effects on the regulation of certain enzymes in the kidney and the thyroid gland [79]. The effects of lithium on the development and maintenance of diabetes-insipidus-like conditions are also known to have significant effects on the levels of vasopressin mRNA in the supraoptic hypothalamic and paraventricular nuclei [80].

Lithium is known to have various metabolic, neurological, and cognitive side effects. Neurotoxicity is a significant factor that limits its use. The presence of lithium poisoning can trigger the accumulation of iron in the brain due to the disruption of the tau cascade. This process hinders the iron efflux from the brain cells and increases the hydroxyl radicals produced by the iron. Lithium has been implicated in various outcomes, such as its effect on the retention of fluids in the hypothalamus [81].

Patients taking lithium are prone to experiencing various types of tremors. These are usually triggered by the initiation or up-titration of the drug’s lithium dose. The usual therapy for patients with this condition is to reduce their lithium dosage to around 0.8 to 1.2 mg/L. This is considered as a fine postural tremor with a higher frequency than other types of tremors. It can also be characterized as an irregular, non-rhythmic tremor of the lower extremities. In some cases, abnormal movement abnormalities such as dysarthria, aphasia, and choreoathetosis can also be caused by severe lithium intoxication [82].

Studies that investigated the effects of lithium on cognitive outcomes revealed that prolonged lithium use was associated with small cognitive impairments [83]. These findings support the idea that prolonged lithium use can affect the development of mental abilities. However, these studies did not find a difference between the short and long-term users of lithium. The effects of lithium on learning and verbal memory were also investigated. The results of the meta-analysis revealed that prolonged lithium use was associated with small cognitive impairments. One study revealed that lithium responders scored higher on a measure of executive function known as the Wisconsin Card Sorting Test [84]. Another study found that lithium use did not affect the cognitive functioning of individuals within two years [20]. However, it did suggest that the effects of lithium on verbal learning could be beneficial. A third study revealed that continued lithium use could reduce the incidence of dementia [85]. The rate of dementia was lower after patients started taking lithium. Lithium chloride can improve spatial learning and memory by increasing GSK3 and Akt phosphorylation [74]. 

In response to the monoamine hypothesis, which hypothesizes that depression is caused by a lack of dopaminergic signaling pathways, studies have been conducted to identify the potential effects of lithium on these pathways. In a study, the authors noted that chronic lithium use could reduce the release of potassium-mediated dopamine [86]. This suggests that the effects of lithium on this pathway could explain the mood-stabilizing effect of its use. In another study, the researchers found that the treatment of cells with lithium prevented Ca^2+^-influx. They believe that the drug’s effect could be mediated by the downregulation of the subunit phosphorylation [87].

Further downstream, treatment with lithium also decreases the activation of calpain, which is a Ca^2+^-dependent protein in the apoptotic cascade. Given its neuroprotective properties, researchers conducted a study on the effects of lithium on the functioning of the brain in a model of ischemic conditions. They found that the rats that were treated with lithium before the middle cerebral artery occlusion had better functional outcomes [88].

In studies on animal models, the effects of lithium on the release of glutamate were observed [89]. These findings suggest that excitotoxicity could be one of the mechanisms by which lithium can cause adverse effects. On the other hand, the effects of lithium on GABAergic signaling are also known to affect the release of glutamate and dopamine [90]. In the brains of rats, chronic lithium use significantly upregulated the levels of glutamic acid decarboxylase and gamma globulin (GABA) and decreased the concentration of dopamine [86]. In addition, acute lithium use significantly increased the levels of this substance in the central nervous system [47].

In downstream signaling, lithium changes the activity of adenylyl cyclase, which is a transcription factor that is involved in regulating the expression of Bcl-2 and BDNF [91]. Long-term lithium use can decrease the activity of PKC [47]. It has also been shown that the drug can decrease the expression of myristoyl alanine-rich C-kinase substrates [92]. It has also been shown that chronic lithium use can increase the levels of Bcl-2. These findings suggest that the drug’s effects could be related to the development of functional outcomes. In clinical and preclinical models, the effects of lithium on the levels of BDNF were observed [93].

It has been theorized that the effects of lithium on the activity of PKC could be related to the treatment of mania [88,94]. In addition, it has been shown that the drug can also decrease the activity of the glycogen synthase kinase-3β [95]. This activity is regulated by direct competition with the binding of Akt and Mg^2+^ [96]. The activation of the Akt pathway is a critical factor that contributes to the survival of neurons following a stroke. It can be triggered by the non-phosphorylation of the GSK3 protein, which leads to its degradation [96]. This process is also triggered by the degradation of the β-catenin substrate. The dual effects of lithium are related to the different functions of the IPPase [96]. For instance, it can directly inhibit the activity of the drug as a competitive inhibitor of the potassium channel, or it can indirectly affect the activity of the drug through the promotion of its phosphorylation [96]. The Akt pathway is not always involved in the neuroprotective effects of certain neuroprotectants [97]. Other studies have shown that it can promote the production of Bcl-2 and other protein components. It can also prevent the activation of p53 and suppress the inflammatory factors c-jun and c-fos [98]. In addition to activating the pathway, lithium can also reduce the size of the infarct by inhibiting the activity of other cellular signaling pathways [99]. 

It was found that the volume of grey matter in the paralimbic and cingulate cortex was significantly higher in the lithium-treated group compared to the control group [100]. In another study, the researchers found that the volume of grey matter in the subgenual cingulate gyrus and the anterior postcentral gyrus was significantly higher in the lithium-treated group compared to the control group [101]. Hajek and colleagues found that the volume of hippocampal volumes was significantly higher in the patients who were treated with lithium [102]. A study conducted by Benedetti and colleagues revealed that the use of lithium was associated with the volume of grey matter from the frontal lobe [103]. A meta-analysis conducted by Bora et al., revealed that the prevalence of lithium users was associated with the volume of grey matter in the anterior cingulate cortex [104].

The proliferative processes in neurons that culminate in observable changes to the volume of grey matter following a stroke can be triggered by the presence of cerebral infarction [105]. In a study, the presence of this type of injury was associated with a decrease in the volume of grey matter [19]. Other studies also suggest a negative correlation between the global grey matter volume and white matter lesions [106]. Due to the effects of lithium on the development of neurogenesis and the remodeling pathways involved in the spontaneous recovery following a stroke, it is possible that the effects of lithium on the volume of grey matter can be observed in a post-stroke population [107].

The effects of lithium on the global volume of grey matter were analyzed in patients. The patients who were given at least 300 mg of lithium daily had a positive change in their grey matter volume. Following a stroke, patients usually experience changes in their mood and cognition, such as post-stroke depression and frontal vascular impairment. Kempton and colleagues also noted that the prevalence of lithium users was associated with the volume of grey matter in a population [108]. A meta-analysis conducted by the researchers revealed that the use of lithium was associated with the global volume of grey matter [101]. Moore and colleagues also revealed that the use of lithium was associated with the cortex grey matter volume [109]. These studies also noted that the effects of lithium on the development of functional outcomes were observed. In healthy individuals, the use of lithium was associated with an increase in regional grey matter volumes [110].

The activity of numerous enzymes, such as the adenylate cyclase and the GSK-3 level is directly affected by the availability of cytosolic magnesium [111]. However, if lithium is present, these same enzymes might be driven to overcompensate by the presence of this substance. For instance, if the level of lithium is high, these enzymes might perform a large corrective response. The members of the lithium-sensitive and magnesium-dependent phosphatases, such as the IPPase and the IMPase, have the potential to form subtle components of a low-Mg failsafe system. In effect, these enzymes are part of a system that can activate different types of failsafe systems. These include the multiple sub-mechanisms that are known to enhance the activity of the inositol phosphates and other related sugar phosphates. It is believed that these two components can help maintain the cell’s energy levels by regulating the interaction between different cellular functions [112].

One of the most common types of sequelae following an ischemic stroke is cognitive impairment. Another common clinical outcome following a stroke is depression. Depression has been known to negatively affect the quality of life of stroke survivors, and it can also lead to disability and worse functional outcomes [107].

Although many stroke survivors experience spontaneous recovery, this doesn’t necessarily mean that they have recovered their pre-stroke functions. It can be mediated by the recovery of compensatory behaviors and baseline functions. Although lithium is widely regarded as a mood stabilizer, its effects on cognition are not known to be definitive. The use of lithium can also negatively affect the cognitive performance of individuals. These effects may be transient and may be affected by the underlying cognitive changes caused by the disease state. Although it has been known that lithium can improve the symptoms of depression, it is not yet clear if its effects on cognitive performance can also be beneficial [107].

Lithium poisonings are most often associated with drug interactions or infections. Patients who are being treated with lithium may develop polynucleosis, which was first observed during their biological exam. The treatment of lithium poisoning is similar to that of cytotoxin. One of the most common complications of lithium exposure is polyuria. Lithium intoxication can also lead to the development of diabetes insipidus. This condition can cause dehydration and increase lithium retention. GSk3β is known to increase the risk of developing diabetes insipidus [113]. 

The effects of GSk3β inhibition on hair growth are also known to be associated with the development of alopecia. Although this complication is not considered a life-threatening condition, it can still have a negative impact on the quality of life of the patient [113].

## 6. Conclusions

The therapeutic effects of lithium are indirectly related to its ability to activate a cascade of fail-safe pathways, designed to protect cells from a variety of threats such as posttraumatic brain injury and depletion of the triphosphate level. Lithium mimics the lowered level of the cellular environment in a way that makes it incredibly effective at protecting cells. These systems are designed to protect neuronal cells from conditions that are detrimental to their survival, such as low levels of magnesium. These systems would be able to regulate, limit, and restore the functions of neurons. Being able to use these systems allows lithium to exert its therapeutic effects. Acting through competition with the magnesium, Li can mimic the effects of low levels of Mg.

## Figures and Tables

**Figure 1 life-12-01680-f001:**
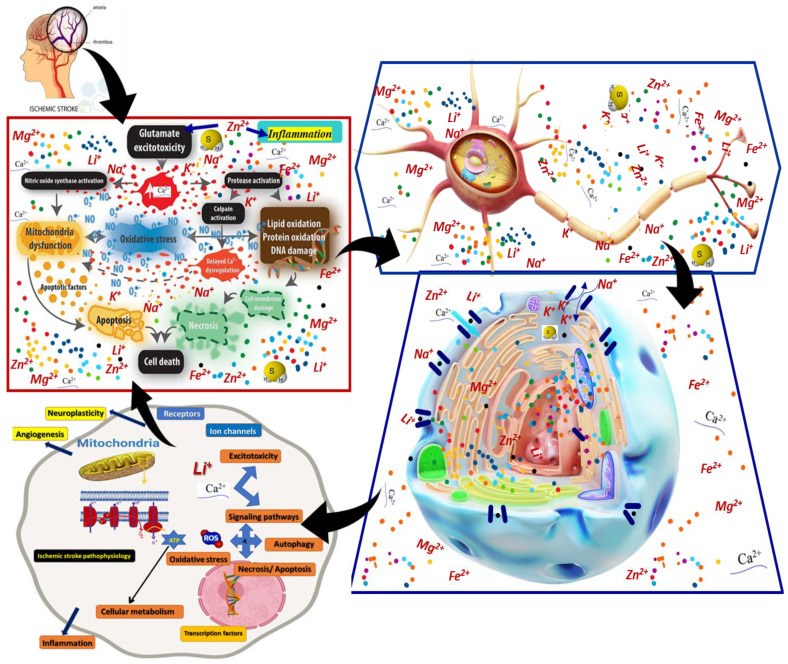
Ionic microenvironment and bio-molecular influenced processes in stroke. Successive “zoom images” (as follows the black arrows) are presented, starting from the tissular representation view to the intra-cellular level of the ionic microenvironment, pointing to the main biological processes influenced by lithium. The Li^+^ dynamics are controlled by various subcellular localized channels and pump proteins.

## Data Availability

Not applicable.

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
