# Peer review of "Lithium Biological Action Mechanisms after Ischemic Stroke"

_life, 2022, doi:10.3390/life12111680_

Round 1

Reviewer 1 Report

Stroke is second cause of dead and a leading cause of disability in surviving patients, with motor sequels, but also with cognitive impairment and post-stroke depression.

This is an interesting paper for regarding potential Lithium neuroprotective role after stroke.

The authors reviewed all potential lithium mechanisms of actions and fiziopathological interactions at different levels, based on vitro and vivo studies.

In the lithium treatment role in depression is well known, and  on post-stroke depression is explicable, there are not yet cert arguments in its role in preventing post-stroke cognitive impairment, also putative mechanisms were listed.

I suggest for the authors a more systematic presentation of  lithium mechanism, avoiding redundancy, for a better ease for the reader

Author Response

Dear reviewer,

Thank you for your constructive evaluation of our manuscript and for a very helpful suggestion of a more systematic presentation of lithium mechanisms which we will include in the Introduction section of the article, as advised. 

We hope to reach the aim of our article by exploring the available data, in vitro and in vivo, regarding lithium intervention mechanisms in stroke patients' recovery.

Reviewer 2 Report

In the review paper, the authors aim to describe the biological actions of lithium after ischemic stroke. A strength of the paper is that it adequately describes the effects of lithium on neurotransmitter-mediated receptor signaling, ion transport, signaling cascades, hormonal regulation, circadian rhythm, and gene expression. Lithium has multifactorial mechanisms of action, and the paper sheds light on these points.

General concept comments

Although the authors mention in the abstract (line 26) that lithium was used at inadequate concentrations, they do not elaborate how the effects of lithium are different at various concentrations.

The authors need to address the side effects of lithium in more detail and also mention the potential drug-interactions.

Specific comments

·         An appropriate and brief description of the biological actions of lithium after ischemic stroke should be included in the introduction section.

·         Line 96 – mention full form of abbreviations (VEGF)

·         Line 117 – rephrase the statement. Mechanical thrombectomy is also an approved treatment option for ischemic stroke. 

·         Line 145 – mention full form of abbreviation (BNF) and add citation

·         Line 413-414. – provide citation for the statement and also mention the detrimental effects of immune activation after ischemic stroke. “Activation of various immune cells, such as B cells, T cells, and macrophages, can help in reducing the damage caused by an ischemic stroke.”

·         Lines 528-533 – the following statement is repeated 3 times. “In another study, the researchers found that the treatment of cells with lithium prevented the Ca2+-influx.”

·         Line 548 – correct to “adenylyl cyclase”.

·         There are many instances throughout the paper where references are missing. Please revise the article and provide citations wherever necessary. Examples of missing references include:

o   Line 106, “The reduction in the 105 volume of grey matter is known to increase the risk of cognitive impairment and dementia.”

o   Line 126, “protect against the development of neuronal apoptosis, and increase the global volume of grey matter.”

o   Line 138 – “The body temperature of patients following an ischemic brain stroke can increase significantly following 6 to 72 hours.”

o   Line 197 – “Studies have shown that lithium can protect nerve cells from multiple attacks.”

o   Line 245 – “Studies have shown that chronic lithium treatment at therapeutic doses provided a complete protection against the effects of glutamate-induced excitotoxicity in various types of neurons.”

o   Lines 251-257, 262

o   Lines 279-281 – “In a mouse model of striatal injury, lithium exposure induced neurogenesis by increasing the number of neurons and their phenotypes. However, in older rats, the effects of lithium on the development of neurogenesis were not as significant.

o   Line 299 – “In studies on mice, lithium was shown to reduce the levels of amyloid precursor protein (APP) in the brains following injury.”

o   Lines 301-303 – “In studies on the effects of lithium on spatial learning and memory, it was discovered that it can reduce the volume of hippocampal volume loss caused by brain injury.”

o   Lines 327-328 – “In a study, lithium was shown to reduce the accumulation of atherosclerotic plaque in mice.”

o   Lines 428-435, 507-509, 516-520

o   Lines 525-526 - “In a study, the authors noted that chronic lithium use could reduce the release of potassium-mediated dopamine.”

o   Lines 540, 550-552, 563-565, 566-568, 571-574, 596-598, 561

·         The articles citied do no match the statements. Please correct the statements in the following lines.

o   Lines 575-577

o   Lines 578-580

o   Lines 593-595

·         There are many occasions with inappropriate self-citations. See examples below.

o   Lines 274-276, 289-291, 323-325, 500-501

·         Figure 1 – The text in the lower left panel of the figure is not clear and the image is of low quality.

·         Figure 2 – Figure legends do not describe the figure. The same figure legend for figure 1 is repeated for figure 2.

Author Response

Dear reviewer,

Thank you very much for your time in providing a detailed, constructive, and very helpful evaluation for improving our manuscript. All your suggestions are considered extremely pertinent in order to increase the scientific accuracy level by following them. Please find our responses below:

General concept comments/answers

Regarding the different effects of lithium at various concentrations, we introduced new paragraphs in the Discussion, addressing the side effects of lithium in more detail and also mentioning the potential drug interactions.

Lines 530-535: “Lithium is known to have various metabolic, neurological, and cognitive side effects. Neurotoxicity is a significant factor that limits its use. The presence of lithium poisoning can trigger the accumulation of iron in the brain due to the disruption of the tau cascade. This process hinders the iron efflux from the brain cells and increases the hydroxyl radicals produced by the iron. Lithium has been implicated in various causes, such as its effect on the retention of fluids in the hypothalamus (81). ”

Lines 536-542: “Patients taking lithium are prone to experiencing various types of tremors. These are usually triggered by the initiation or up-titration of the drug's lithium dose. The usual therapy for patients with this condition is to reduce their lithium dosage to around 0.8 to 1.2 mg/L. This is considered as a fine postural tremor with a higher frequency than other types of tremors. It can also be characterized as an irregular, non-rrhythmic tremor of the lower extremities. In some cases, abnormal movement abnormalities such as dysarthria, aphasia, and choreoathetosis can also be caused by severe lithium intoxication (82). ”

Lines 664-670: “Lithium poisonings are most often associated with drug interactions or infections. Patients who are being treated with lithium may develop polynucleosis, which was first observed during their biological exam. The treatment of lithium poisoning is similar to that of cytotoxin. One of the most common complications of lithium exposure is polyuria. Lithium intoxication can also lead to the development of diabetes insipidus. This condition can cause dehydration and increase lithium retention. GSk3β is known to increase the risk of developing diabetes insipidus (113). ”

Lines 671-673: “The effects of GSk3β inhibition on hair growth are also known to be associated with the development of alopecia. Although this complication is not considered a life-threatening condition, it can still have a negative impact on the quality of life of the patient (113). ”

Specific comments/answers

An appropriate and brief description of the biological actions of lithium after an ischemic stroke was included in the introduction section: lines 82 - 100

Line 96 – mention full form of abbreviations (VEGF) – new line 115: vascular endothelial growth factor

Line 117 – rephrase the statement. Mechanical thrombectomy is also an approved treatment option for ischemic stroke. 

New line 137, ref. 22,23

Line 145 – mention full form of abbreviation (BNF) and add citation

New line 165 – was corrected the error, is about BDNF, ref. 36

Line 413-414. – provide citation for the statement and also mention the detrimental effects of immune activation after ischemic stroke. “Activation of various immune cells, such as B cells, T cells, and macrophages, can help in reducing the damage caused by an ischemic stroke.” – new lines 433 – 436, ref. 65, “The inflammatory response to an acute cerebral infarction is a dual mechanism, detrimental, but also a protective factor that can determine the outcome of the stroke. Activation of various immune cells, such as B cells, T cells, and macrophages, can help in reducing the damage caused by an ischemic stroke (65).

Lines 528-533 – the following statement is repeated 3 times. “In another study, the researchers found that the treatment of cells with lithium prevented the Ca2+-influx.” – new lines 566 – 571, the redundant paragraphs were deleted.

Line 548 – correct to “adenylyl cyclase”. New line 585, the correction was made

There are many instances throughout the paper where references are missing. Please revise the article and provide citations wherever necessary. Examples of missing references include:

o   Line 106, “The reduction in the 105 volume of grey matter is known to increase the risk of cognitive impairment and dementia.”

New line: 125, ref. 19

o   Line 126, “protect against the development of neuronal apoptosis, and increase the global volume of grey matter.”

New line: 146, ref. 30

o   Line 138 – “The body temperature of patients following an ischemic brain stroke can increase significantly following 6 to 72 hours.”

New line: 159, ref. 34

o   Line 197 – “Studies have shown that lithium can protect nerve cells from multiple attacks.”

New line: 218,  “Studies have shown that lithium can protect nerve cells from multiple attacks, as toxins, stress, ischemia, and injury (41). ”

o   Line 245 – “Studies have shown that chronic lithium treatment at therapeutic doses provided a complete protection against the effects of glutamate-induced excitotoxicity in various types of neurons.”

New line: 265, ref. 47,48

o   Lines 251-257, 262

New lines: 271 – 279 – paragraph changed, eliminating part of it. Ref. 49,50

New line 283, ref. 51

o   Lines 279-281 – “In a mouse model of striatal injury, lithium exposure induced neurogenesis by increasing the number of neurons and their phenotypes. However, in older rats, the effects of lithium on the development of neurogenesis were not as significant.

New lines: 299-302, ref.9

o   Line 299 – “In studies on mice, lithium was shown to reduce the levels of amyloid precursor protein (APP) in the brains following injury.”

New line: 315, ref. 53

o   Lines 301-303 – “In studies on the effects of lithium on spatial learning and memory, it was discovered that it can reduce the volume of hippocampal volume loss caused by brain injury.”

New lines: 323 – 325, ref. 56

o   Lines 327-328 – “In a study, lithium was shown to reduce the accumulation of atherosclerotic plaque in mice.”

New lines: 348-349, ref. 59

o   Lines 428-435, 507-509, 516-520

New lines: 450 – 457, ref. 60,70,71

New lines: 543 – 545 , ref. 83, Studies that investigated the effects of lithium on cognitive outcomes revealed that pro-longed lithium use was associated with small cognitive impairments (83).

New lines: 552 – 553, ref. 84, One study revealed that lithium responders scored higher on a measure of executive func-tion known as the Wisconsin Card Sorting Test (84).

o   Lines 525-526 - “In a study, the authors noted that chronic lithium use could reduce the release of potassium-mediated dopamine.”

New lines: 562-563, ref. 86

o   Lines 540, 550-552, 563-565, 566-568, 571-574, 596-598, 561

New lines: 577, ref. 89

New lines: 585-587, ref. 91

New lines: 600-604, ref. 97

New lines: 604-606, ref. 98

New lines: 609-610, ref.100

New lines: 634-635, ref.109, paragraph changed

New lines: 599, ref. 96

The articles citied do no match the statements. Please correct the statements in the following lines.

o   Lines 575-577

New lines: 613-615, ref.101

o   Lines 578-580

New lines: 617-618, ref.104

o   Lines 593-595

New lines: 631-632, ref.108

There are many occasions with inappropriate self-citations. See examples below.

  • Lines 274-276, - New lines: 295-298, changed - ref.9
  • Lines 289-291, - New lines: 310-312, changed -ref.52
  • Lines 323-325, - New lines: 345-347, changed -ref.58
  • Lines 500-501, - New lines: 523-524, changed -ref.78

Figure 1 – The text in the lower left panel of the figure is not clear and the image is of low quality. – the figure was optimized for a better quality

Figure 2 – Figure legends do not describe the figure. The same figure legend for figure 1 is repeated for figure 2. – the error was corrected and a new legend was introduced.

Round 2

Reviewer 2 Report

In the review paper, the authors aim to describe the biological actions of lithium after ischemic stroke. A strength of the paper is that it adequately describes the effects of lithium on neurotransmitter-mediated receptor signaling, ion transport, signaling cascades, hormonal regulation, circadian rhythm, and gene expression. Lithium has multifactorial mechanisms of action, and the paper sheds light on these points.

Thank you for making the changes.

Please fix lines 631-632, “Kempton and colleagues also noted that the prevalence of lithium users was associated with the volume of grey matter in a population (108).” The article cited is by Machado-Vieira. 

Author Response

Dear reviewer,

Thank you very much for your time and for your help. 

lines 631-632, “Kempton and colleagues also noted that the prevalence of lithium users was associated with the volume of grey matter in a population (108).” The article cited is by Machado-Vieira - it was an error in my Mendeley selection of the article. Now the new reference is:

  1. Kempton MJ, Geddes JR, Ettinger U, Williams SCR, Grasby PM. Meta-analysis, database, and meta-regression of 98 structural imaging studies in bipolar disorder. Arch Gen Psychiatry. 2008;65(9):1017–32.

With the greatest consideration! Thank you very much!